# L-Aspartate and L-Glutamine Inhibit Beta-Aminobutyric Acid-Induced Resistance in Tomatoes

**DOI:** 10.3390/plants11212908

**Published:** 2022-10-29

**Authors:** Adam Janotík, Kateřina Dadáková, Jan Lochman, Martina Zapletalová

**Affiliations:** Faculty of Science, Masaryk University, Kotlarska 2, 611 37 Brno, Czech Republic

**Keywords:** β-aminobutyric acid, BABA-induced resistance, *Pseudomonas syringae*, jasmonic acid, amino acids

## Abstract

Plant diseases caused by pathogens lead to economic and agricultural losses, while plant resistance is defined by robustness and timing of defence response. Exposure to microbial-associated molecular patterns or specific chemical compounds can promote plants into a primed state with more robust defence responses. β-aminobutyric acid (BABA) is an endogenous stress metabolite that induces resistance, thereby protecting various plants’ diverse stresses by induction of non-canonical activity after binding into aspartyl-tRNA synthetase (AspRS). In this study, by integrating BABA-induced changes in selected metabolites and transcript data, we describe the molecular processes involved in BABA-induced resistance (BABA-IR) in tomatoes. BABA significantly restricted the growth of the pathogens *P. syringae* pv. tomato DC3000 and was related to the accumulation of transcripts for pathogenesis-related proteins and jasmonic acid signalling but not salicylic acid signalling in Arabidopsis. The resistance was considerably reduced by applying amino acids L-Asp and L-Gln when L-Gln prevents general amino acid inhibition in plants. Analysis of amino acid changes suggests that BABA-IR inhibition by L-Asp is due to its rapid metabolisation to L-Gln and not its competition with BABA for the aspartyl-tRNA synthetase (AspRS) binding site. Our results showed differences between the effect of BABA on tomatoes and other model plants. They highlighted the importance of comparative studies between plants of agronomic interest subjected to treatment with BABA.

## 1. Introduction

Due to the inability to move, plants are exposed to various environmental stresses, both biological and abiotic, without having the possibility to escape. As a result of numerous pathogens and pest attacks, plants have formed passive and active physical and biochemical barriers. Once innate immunity is insufficient, local and systemic inducible responses can be activated to fortify defences. Defence responses are activated upon recognising the pathogen when the robustness of early activation and response cannot prevent the pathogen’s spread. Often, plant defence mechanisms are ineffective, leading to extensive plant infections resulting in catastrophic harvest failures and causing considerable economic and social problems worldwide. 

Even though plants do not possess mobile immune system cells, defence responses can be activated at both the local and systemic levels after pathogen recognition. In these cases, the timing and strength of the immune response are decisive. The state of induced resistance is then characterised by a faster and more potent immune response of the plant activated upon the pathogen perception. Induced resistance can be activated upon the stimulus of different origins and provides resistance against a broad spectrum of stresses either through systemic acquired resistance (SAR) or induced systemic resistance (ISR). Besides native intrinsic activation, induced resistance can be activated by exogenously applied chemical compounds. Today, managing plant defences is a promising approach, especially using priming agents that prepare plants to enhance their defence responses. Defence priming causes enhanced expression of genes related to stress and defence [1], including many defence regulatory transcription factors [2]. Defence priming is now considered an essential component of various types of systemic plant immunity. 

One of the most influential and promising defence priming agents is the non-proteinaceous amino acid beta-aminobutyric acid (BABA), which induces broad-spectrum resistance to pathogens covering the oomycetes, fungi, bacteria, nematodes, and insects in many plant species [3]. BABA was considered a xenobiotic compound, but recent research showed that it accumulates in plants exposed to stress [4] and acts as an endogenous stress metabolite [5]. The molecular mechanism of BABA-induced resistance (BABA-IR) was partially elucidated in Arabidopsis plants via the binding of R-BABA to the aspartate binding site of the aspartyl-tRNA synthetase (AspRS) IBI-1 and induction of its non-canonical activity resulting in translocation to the cytoplasm [6]. Here, VOZ1 and VOZ2 transcription factors were identified as IBI1-interacting partners, and this interaction repressed the expression of ABA genes resulting in increased expression of pattern-triggered immunity (PTI) genes and callose-associated defence. An increased Asp level after R-BABA application further supported the proposed mechanism; however, direct evidence of the inhibition of the BABA effect on IBI-1 activity is missing. Interestingly, BABA-IR inhibition was observed in Arabidopsis after the L-glutamine treatment, which prevents a general amino acid stress inhibition in plants [7].

Tomato (*Solanum lycopersicum*) is one of the most important agricultural plants in the world, with an annual production of 186 million tons in 2020 [8]. Besides its agricultural importance, the tomato is widely used in research due to the presence of features such as compound leaves, fleshy fruit, and sympodial shoot. In contrast, other model plants, such as Arabidopsis and rice, lack these agronomically essential traits. In addition, the tomato is a member of the Solanaceae family and studies conducted on the tomato can be easily applied to members of this family, which also includes commercially important crops such as tobacco, potatoes, peppers, and eggplants, making it a vital research material [9]. Previous studies showed that BABA induced resistance to biotrophic and hemibiotrophic pathogens in tomato plants, but our study further advances our understanding of the detailed description of the molecular mechanism [3]. 

Here, we investigated the effect of BABA treatment in combination with amino acids Asp and Gln on BABA-IR in tomatoes. We used cultivars *Solanum lycopersicum* cv. Micro-Tom and cv. Amateur, representing a model tomato cultivar and a widely grown tomato cultivar, respectively, and the established virulent bacterial model pathogen *Pseudomonas syringae* pv. tomato DC3000. Analysis of resistance in conjunction with analysis of amino acid changes, transcripts, and phytohormones involved in resistance demonstrate that treatment with Asp and Gln reduces the rate of BABA-IR, which in tomato, appears to be mediated by JA rather than the SA signalling pathway.

## 2. Results

### 2.1. L-Glutamine and L-Aspartate Reduce the BABA-IR in Tomato

To determine the protective effect of BABA treatment against the pathogen *P. syringae* pv. tomato DC3000, we treated 6–7-week-old *S. lycopersicum* cv. MicroTom and cv. Amateur leaflets with 5 mM BABA, L-Asp, or equimolar mixtures BABA:Asp and BABA:Gln through petiole aspiration to avoid the formation of small necrotic lesions observed after BABA treatment by spraying [10,11]. In the case of *S. lycopersicum* cv. MicroTom, bacterial inoculation was performed by spraying the bacterial suspension on the leaflets or by direct syringe infiltration of bacteria to the apoplast of the leaflets 24 h after BABA treatment. Previous studies with diverse fungi and bacteria showed that BABA has no direct toxic effect. Thus, BABA-mediated resistance is most likely based on the activation of host resistance mechanisms [12]. The infection was evaluated 48 h later by the previously described qPCR method [13] using primers for the *P6* gene (Solyc00g174340) from tomato and the *PSPTO_4345* gene (GenBank: CP047072) for *P. syringae* pv. tomato DC3000. Noticeably, the resistance depended on the technique of the application. The application by syringe did not induce significant resistance after the BABA or Asp treatment (Figure 1A), but application by spraying induced considerable resistance after the BABA treatment (Figure 1B). Moreover, treatment with an equimolar mixture of BABA: Asp weakened this resistance to the basal level (Figure 1B).

For the results obtained in the case of the *S**. lycopersicum* cv. Amateur, we inoculated the pathogen only using the spraying technique. Compared with the MicroTom cultivar, we also used 5 mM L-Gln and equimolar mixtures BABA:Gln. Not only BABA but also BABA enriched with amino acids Asp and Gln induced significant resistance in the Amateur cultivar but to a lesser extent (Figure 1C). Noticeably, treatment with an equimolar mixture of BABA:Asp or BABA:Gln resulted in the significant weakening of observed BABA-IR (Figure 1C).

### 2.2. BABA Did Not Induce Changes in L-Aspartate Levels in Tomato

The result from the resistance experiments indicated the role of aspartic acid and glutamine on the level of BABA-IR. Hence, a detailed amino acids analysis was carried out in *S. lycopersicum* cv. MicroTom leaflets treated with 5 mM solutions of BABA, L-Asp, or equimolar mixture BABA:Asp and *S**. lycopersicum* cv. Amateur leaflets treated with 5 mM solutions of BABA, L-Asp, L-Gln, or equimolar mixtures BABA:Asp and BABA:Gln through petiole aspiration. In all BABA-treated samples, we found a significantly increased level of BABA corresponding to the basal level of Gln, which greatly exceeded the basal level of aspartate (Figure 2A,B).

In contrast to previous data on *Arabidopsis* plants [15], BABA treatment of both tomato cultivars through petiole aspiration did not induce a significant increase in L-Asp level (Figure 2A), while in the case of *S**. lycopersicum* cv. Amateur, we even detected a decrease in the L-Asp level (Figure 2B). This result could be a consequence of metabolic disbalance in susceptible cultivars after BABA treatment because we observed a decrease in many other amino acids such as Glu, Asn, Ala, Trp, Phe, or Lys. This phenomenon will need to be confirmed and described in more detail in further studies.

Surprisingly, in both tomato cultivars, the treatment with Asp did not result in its significantly elevated level but we observed a substantial increase in the level of Gln together with Asn, the common nitrogen carriers playing the primary role in the recycling, storage, and transport of nitrogen in plants (Figure 2A,B) [16,17]. In the case of treatment with equimolar mixtures of BABA:Asp, we observed an additive effect of both substances. In *S**. lycopersicum* cv. MicroTom, treatment resulted in elevated levels of aromatic amino acids Tyr, Trp, and Phe and amino acids Val and Lys, similar to BABA and Asp treatment. *S**. lycopersicum* cv. Amateur treatment decreased Glu and Thr levels, identical to the BABA treatment (Figure 2A,B). In both tomato cultivars, we observed a significant increase in Gln level similar to the Asp treatment (Figure 2A,B). The treatment of *S**. lycopersicum* cv. Amateur plants with Gln and equimolar mixtures BABA:Gln led to a significant increase in Asn, Gln, and Tyr levels when, in the case of the BABA:Gln treatment, we found increased levels of the aromatic amino acids Trp and Phe (Figure 2B). 

### 2.3. BABA up-Regulated Transcripts and Signalling in the Tomato 

To demonstrate the role of defence genes in the resistance of *S**. lycopersicum* cv. Amateur leaflets induced by BABA and amino acids treatment, we measured the expression of four defence genes, *P4* (PR1 protein), *HEV2* (hevein 2), *NP24* (osmotin 1)*,* and *ACS2* (1-aminocyclopropane-1-carboxylic acid synthase 2), whose increased expression was previously demonstrated in BABA-IR [11]. The biological function of gene *P4* is unclear, *HEV2* gene coding protein acting as a chitinase, *NP24* gene coding thaumatin-like protein PR-NP24, and *ACS2* gene coding enzyme e ACC synthase regulating the ethylene (ET) synthesis [18].

The transcript levels were analysed in leaves collected 24 h post-treatment through petiole aspiration with 5 mM solutions of BABA, L-Asp, L-Gln, or equimolar mixtures BABA:Asp and BABA:Gln and two days after infection with *P. syringae* pv. tomato DC3000. In all treatments except the Asp treatment, we observed a strong up-regulation of the P4 transcript after 24 h. Interestingly, the Asp treatment did not up-regulate any other measured genes when only the tBABA and BABA:Asp treatment up-regulated the HEV2 transcript and down-regulated the ACS2 transcript. Finally, the NP24 transcript was up-regulated only in the BABA:Asp treatment (Figure 3A).

On the other hand, two days after infection, we observed strong up-regulation of all measured genes in all treatment conditions and inoculated control leaflets treated with MgCl_2_ buffer containing 0.02% Silwet L77 as a mock inoculation (Figure 3B). Two days after inoculating leaflets with the pathogen *P. syringae*, compared to the inoculated control, we observed a significantly higher *ACS2* transcript accumulation, coding a key regulatory enzyme in the ET synthesis pathway in all treatments (Figure 3B). Moreover, the P4 transcript was significantly accumulated in all treatments except BABA and the NP24 transcript was significantly increased only in the L-Gln treatment (Figure 3B). 

BABA induces resistance via several hormones, including SA [19], JA [20], abscisic acid [21] or ET [12]. In both potato and Arabidopsis, BABA potentiates SA-dependent defence against pathogens [2]. Indeed, in Arabidopsis, BABA-induced priming seems to be also ABA-dependent [6]. Surprisingly, here in *S**. lycopersicum* cv. Amateur leaflets, BABA treatment alone or in combination with amino acids L-Asp and L-Gln led to a significant decrease in SA compared to its higher level after L-Asp treatment alone (Figure 4A). In addition, the application of BABA to tomato leaflets did not change the level of ABA and its combination with amino acids L-Asp and L-Glu reduced the amplitude of the increase in the ABA level caused by the amino acids themselves. On the other hand, treatment with 5 mM solutions of BABA, L-Asp, L-Gln, or equimolar mixtures BABA:Asp and BABA:Gln significantly induced the accumulation of JA and its endogenous bioactive conjugate JA-Ile but to a lesser extent (Figure 4B). Consistent with the transcriptomic data for the ACS2 enzyme, one of the key enzymes of the ET synthesis pathway in tomato and Arabidopsis [11,22], we did not observe an increase in ET concentration in leaflets treated with 5 mM BABA through petiole aspiration, but only after the spray treatment with 5 mM BABA, as described earlier [11] (Figure 4C). 

## 3. Discussion

The phenomena of BABA-IR in different plants were proven by various studies [23,24]. Here, we tested the effectiveness of BABA, L-Asp, and L-Gln or their equimolar mixture treatments on the resistance of the dwarf cultivar *Solanum lycopersicum* cv. Micro-Tom [25] and *S**olanum lycopersicum* cv. Amateur against the pathogenic bacterium *Pseudomonas syringae* pv. tomato DC3000 [26]. *Solanum lycopersicum* cv. Micro-Tom represents a preferred variety for molecular research in tomatoes [27] and *S**olanum lycopersicum* cv. Amateur is a good model system for an agricultural crop. The L-Asp was selected due to the previous finding that aspartyl-tRNA synthetase serves as an R-BABA receptor in *Arabidopsis* and L-Asp accumulates after BABA application [15]. On the other hand, the L-Gln treatment of the *Arabidopsis* plant inhibited BABA-IR previously [7]. 

A BABA-IR against pathogenic bacteria *P. syringae* pv. tomato DC3000 depended on the technique of pathogen application when, in the case of *S. lycopersicum* cv. Micro-Tom, we observed a high basal resistance level to *P. syringae* pv. tomato DC3000 with limited symptoms corresponding with the previous finding [28]. In the case of syringe infiltration, where the bacterial pathogen is delivered directly to the apoplastic space, we observed no induction of resistance compared to the application of bacterial pathogen on leaves. This result suggests that BABA application-mediated priming of tomato defence mechanisms involved in stomata closure restricts pathogen penetration to the apoplast as a part of the pattern-triggered immunity (PTI). PTI controls the production of long-distance signals that systemically prime plants against future attacks, resulting in systemic acquired resistance [29,30,31]. Noticeably, in the case of *S**. lycopersicum* cv. Amateur, the significantly increased resistance, even lower than BABA, was observed after applying 5 mM L-Asp and L-Gln. This finding agrees with the study of Kadotani et al., 2016, in which strong rice resistance against rice blast fungus *Magnaporthe grisea* was observed after applying 10 mM proteinaceous amino acids L-Glu, L-Asn, L-Met, and L-Asp [32]. 

In Arabidopsis thaliana, the molecular mechanism described by Luna et al., 2014 suggests induction of the non-canonical activity of AspRS mediated by the binding of R-BABA to a catalytic site of the enzyme and increase in L-Asp level [15]. Here, this suggestion seems to be supported by the observed effect of L-Asp on reducing BABA-IR resistance. However, the measured level of L-Asp showed no increase 24 h after L-Asp or BABA treatment, and the BABA level in leaflets was approximately 10-times higher compared to L-Asp due to its persistence in plants. On the other hand, we detected an increased level of L-Gln and L-Asn after L-Asp or BABA/L-Asp treatment, indicating its rapid metabolisation to these typical amino acids used by the nitrophilous plants for nitrogen storage [33]. Indeed, we observed a significant reduction of BABA-IR after L-Gln treatment, in agreement with previous studies on Arabidopsis [34]. Therefore, it seems probable that the observed decrease in BABA-IR after L-Asp treatment could be somewhat due to an increased level of L-Gln than due to competition of L-Asp with BABA regarding the binding site. In the case of *S. lycopersicum* cv. Micro-Tom, a significant increase in the level of precursor amino acids of the phenylpropanoid pathway (Phe, Tyr, and Trp) involved in increased resistance was observed [35] after BABA treatment. In addition, an increased level of amino acid Lys, representing a precursor of pipecolic acid taking a role in defence reaction [36], may also play a role in this priming mechanism. Interestingly, in the case of *S. lycopersicum* cv. Amateur, we found an increased level of these amino acids only after BABA/L-Gln treatment, showing different mechanisms operating in tomato cultivars with different resistance mechanisms, as reported earlier [11].

BABA treatment strongly induced the expression of the P4 and HEV2 genes, which corresponds well with our previous study on *S. lycopersicum* cv. Amateur [11]. On the other hand, compared to that study, we did not observe any increase in the NP24 and ACS2 transcripts. We suggest that this discrepancy is a result of a different application method. Applying BABA by foliar spraying caused the formation of hypersensitive reaction-like lesions on the leaves followed by ET synthesis [11,37], which could be related to stress responses triggered by high BABA concentrations in the drying droplets (white deposits) after spraying. This observation corresponds with the well-demonstrated involvement of ET in the regulation of the degree of PCD during plant–pathogen interactions [38], where the initiation of hypersensitive reaction results in a large burst of ET [39]. However, the BABA was aspirated into the leaflet through the petiole, causing no formation of hypersensitive reaction-like lesions and, thus, no ET production. In keeping with this hypothesis, applying high BABA concentrations (>20 mM) to the petiole of detached tomato leaflets induces the formation of HR-like lesions. Pathogen inoculation led to strong up-regulation of all selected genes; however, in the case of BABA pretreatment, there was no visible defence priming effect demonstrated by the more intense expression [40]. Only in the case of the ACS2, the key enzyme of ET synthesis, transcript was significantly higher than the control, supporting the suggested ET-dependent mechanisms in BABA-induced resistance in tomatoes [11,41]. 

In Arabidopsis, BABA treatment was accompanied by the accumulation of SA and caused significant changes in the abundance of isochorismate synthase (ICS), which is directly involved in SA biosynthesis [12,42]. Here, in *S. lycopersicum* cv. Amateur, we observed no accumulation of SA after BABA treatment. On the other hand, treatment of *S. lycopersicum* cv. Amateur with BABA and/or amino acids L-Asp and L-Gln increased the content of JA and JA-Ile in *S. lycopersicum* cv. Amateur leaflets in a manner similar to BABA-IR towards *P. infestans* in the tomato def mutant, which is defective in JA accumulation [43]. Noticeably, BABA treatment reduced the accumulation of abscisic acid (ABA) after L-Asp and L-Gln treatment, which corresponds to the recently observed suppression of ABA-inducible abiotic stress genes during BABA-IR against the biotrophic oomycete *Hyaloperonospora arabidopsidis* [6]. 

This work shows that the signalling processes and immune response activated during BABA-IR in tomatoes differed from those in Arabidopsis in response to *P. syringae* pv. tomato DC3000 in *S. lycopersicum* cv. Amateur. In BABA-treated tomato plants, the defence reaction is controlled by jasmonic acid signalling but not salicylic acid signalling. Moreover, BABA treatment of tomato caused no change in Asp levels; however, Asp- and Gln-treatment of tomato reduced BABA-IR (Figure 5). In conclusion, comparative studies between BABA-treated plants of agronomic interest must be carefully interpreted and expanded to other plant systems in general. 

## 4. Materials and Methods

### 4.1. Plant Material

*Solanum lycopersicum* cv. Micro-Tom and Amateur were grown at 65% humidity, 24 °C with a 16 h daily light period (light intensity 100 µmol/m^2^). The detached leaflets from 6–7-week-old plants were immersed for 24 h in aqueous solutions of 5 mM BABA, 5 mM BABA with 5 mM Asp, 5 mM BABA with 5 mM Gln, 5 mM Asp and 5 mM Gln adjusted to pH = 7. Subsequently, the leaflets were inoculated with the pathogenic bacteria *Pseudomonas syringae* pv. tomato DC3000 and plants were incubated in a climatic chamber at 24 °C for 72 h. Leaves were collected before pathogen inoculation for amino acid and phytohormone level analysis and gene expression analysis and 72 h after the pathogen inoculation for resistance and gene expression analysis.

### 4.2. Plant Resistance Analysis against Pseudomonas Syringae

*P. syringae* pv. tomato DC3000 was inoculated into King’s B medium and cultured on a shaker at 225 RPM at 28 °C for 24 h. Subsequently, the culture was centrifuged at 4000× *g* for 10 min. The bacterial culture was diluted with 10 mM MgCl_2_ to an A_620_ of 0.2, corresponding to 10^8^ CFU/mL concentration. Next, the bacteria were diluted to a concentration of 10^7^ CFU/mL with 10 mM MgCl_2_ containing 0.1% Silwet L77 (AgroBio, Opava, Czechia). This was followed by applying bacteria into the leaves using direct infiltration into the leaf apoplast using a 1 mL needleless syringe or a brush to avoid wounding tissue until both the abaxial and adaxial surfaces were uniformly wet according to the method described previously [44]. The MgCl_2_ buffer containing 0.02% Silwet L77 was applied as a mock inoculation. The leaves were incubated in a climatic chamber for 72 h with water supplied by wet cotton at 23 °C, 85% humidity (16 h photoperiod), and then frozen in liquid nitrogen and stored at −70 °C until DNA isolation. 

DNA was isolated according to the CTAB protocol [45]. Isolated DNA was analysed by qPCR method using Luna^®^ Universal qPCR Master Mix (NEB, Ipswich, MA, USA) in a Light Cycler 480 (Roche, Mannheim, Germany). Primers for *P6* (cv. Amateur) or *Tip41* (cv. Micro-Tom) genes were used for tomato quantification and primers for *PSPTO_4345* gene (GenBank: CP047072) for pathogen quantification. Data were evaluated by the ΔC_t_ method [14]. PCR mixtures were prepared in a total volume of 15 µL and contained 4 µL of template DNA, 7.5 µL of Luna^®^ Universal qPCR Master Mix (2×), and forward and reverse primer (Appendix A) in a final concentration of 0.3 µM. qPCR temperature profile consisted of a 150-sec denaturation step at 95 °C followed by 45 cycles consisting of a 10-sec denaturation step at 95 °C and a 30-sec annealing extension step at 60 °C. 

### 4.3. Amino Acid Levels Quantification

Amino acids were extracted 24 h after the solution treatment (BABA, L-Asp, L-Gln, BABA:Asp, and BABA:Gln). A 1 mL of extraction buffer consisting of 0.1 M HCl with the addition of 4.6 μg/mL 2-aminoadipic acid as an internal standard was added to 100 mg of leaf powder, mixed thoroughly, and incubated on ice for 5 min. Samples were centrifuged for 10 min at 4 °C and 15,000× *g*. A total of 500 μL of supernatant was diluted with 100 μL of methanol. Subsequently, solid-phase extraction was performed using Macherey-Nagel Chromabond^®^ C18, 100 mg of solid-phase per 5 mL colony. The colony was washed with 1 mL of methanol and then with 1 mL of equilibration buffer (20% MeOH in 0.1 M HCl). Samples (600 μL) were applied to the column, and the column was washed with 400 μL of equilibration buffer, resulting in a volume of 1 mL of extracted amino acids. Extracted amino acids were derivatised and analysed, as described previously [46]. 

### 4.4. Transcription of Pathogenesis-Related Genes

Plant material before and after 72 h post-pathogen inoculation was used. Plant RNA was isolated by TRI REAGENT (Merck, Darmstadt, Germany) and treated by RapidOut DNA Removal Kit (Thermo Scientific™, Waltham, MA, USA). Reverse transcriptase reactions were performed with the ImProm-II reverse transcription system (Promega, Madison, WI, USA). According to the manufacturer’s instructions, the obtained cDNA was amplified by qPCR using gene-specific primers (Appendix A) and Luna^®^ Universal qPCR Master Mix (NEB, Hertfordshire, UK). The PCR mixtures were prepared in a total volume of 10 µL and contained 1 µL of template cDNA, 5 µL of Luna^®^ Universal qPCR Master Mix (2×), and forward and reverse primer in a final concentration of 0.3 µM. qPCR temperature profile consisted of a 150-sec denaturation step at 95 °C followed by 45 cycles consisting of a 20-sec denaturation step at 95 °C and a 40-sec annealing extension step at 60 °C. 

### 4.5. Phytohormone Levels Analysis

Phytohormone levels were measured 24 h after solution treatment (5 mM BABA, 5 mM BABA with 5 mM Asp, 5 mM BABA with 5 mM Gln, 5 mM Asp, 5 mM Gln, pH = 7). A total of 50 mg of leaf powder with 20 ng of internal standard (o-anisic acid) were extracted twice into 1 mL of cold 10% methanol with 3 min of sonication and 30 min of shaking at 4 °C and 700 rpm. Samples were centrifuged for 10 min at 4 °C and 15,000× *g*, the supernatants were applied to SPE columns (HLB, Merck, Darmstadt, Germany) washed with 2 mL of water and 1 mL of water, and samples were applied and washed with 1 mL of 10% MeOH and eluted with 2 mL of methanol. The samples were dried, thoroughly resuspended in 40 µL of 0.04% formic acid and 15% acetonitrile, and filtered through 0.22 µm filters. Phytohormone levels were measured by LC–MS/MS (6545, Agilent, Santa Clara, CA, USA). Phytohormones were separated by reverse-phase column (Poroshell 120 SB-C18 2.1 × 100 mm^2^, 2.7 µm, Agilent, Santa Clara, CA, USA) and gradient solution technique. A total of 0.04 % formic acid (A) and acetonitrile (B) were used as mobile phases with the following gradient conditions: 15% B from 0 to 5 min, 15 to 45% B from 5 to 15 min, 45 to 47% B from 15 to 22 min and finally, 100% from 22 to 32 min, at a flow rate of 0.4 mL/min and injection volume of 10 µL. Conditions of electrospray ion source were as follows: acquisition mode 100–1700 *m/z*, gas flow 8 L/min, gas temperature 240 °C, capillary voltage 3000 V, fragmentor voltage 150 V. Positive ion polarity was used for the detection of methyl jasmonate and jasmonoyl–isoleucine and negative for the o-anisic, jasmonic and salicylic acid. The analytes were qualified by comparison of retention times and mass spectra with standards and quantified using molecular ion peak areas. MassHunter Quantitative Analysis software was used for drawing calibration curves.

## Figures and Tables

**Figure 1 plants-11-02908-f001:**
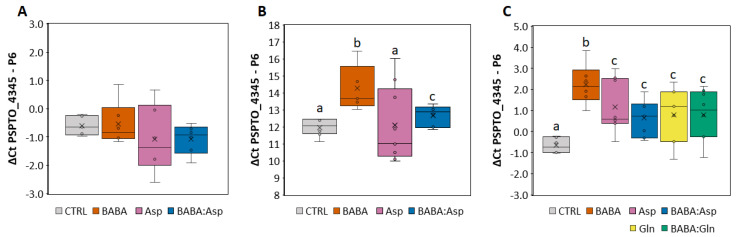
Resistance levels of tomato against pathogen *P. syringae.* (**A**,**B**) *Solanum lycopersicum* cv. MicroTom leaflets were inoculated with *Pseudomonas syringae* pv. tomato DC3000 by direct syringe infiltration of bacteria into the apoplast (**A**) or by spraying the bacterial suspension on the leaflets (**B**) treated for 24 h with 5 mM BABA, Aspartate, or their equimolar mixture (BABA:Asp). (**C**) *Solanum lycopersicum* cv. Amateur leaflets were inoculated with *Pseudomonas syringae* pv. tomato DC3000 by spraying the bacterial suspension on the leaflets treated for 24 h with 5 mM BABA, Aspartate (Asp), Glutamine (Gln), or their equimolar mixture (BABA:Asp, BABA:Gln). The level of resistance was measured after 72 h by qPCR and evaluated by the ΔΔCt method [14] using the *P6* gene (Solyc00g174340) from tomato and the *PSPTO_4345* gene (GenBank: CP047072) from *P.*
*syringae* pv. tomato DC3000. Significant differences compared with the water-treated control were determined using the ANOVA, and Duncan’s test is a post-hoc test; different lowercase letters indicate significant differences at *p* ≤ 0.05 (Duncan’s test).

**Figure 2 plants-11-02908-f002:**
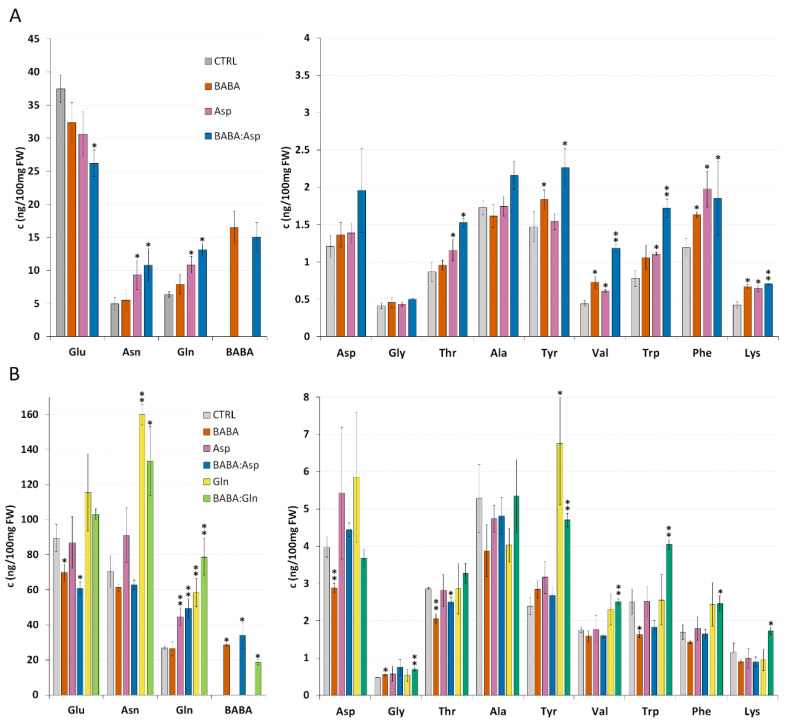
Amino acid levels in tomato plants. Amino acid levels of 8-week-old leaflets of *Solanum lycopersicum* cv. MicroTom (**A**) and *Solanum lycopersicum* cv. Amateur (**B**) plants. Leaflets were treated for 24 h with BABA or amino acids solutions (BABA, L-Asp, BABA: Asp, L-Gln, BABA: Gln), and amino acid levels were measured by the HPLC method. Data are means from free biological replicates for each time; the errors represent standard errors of means. Statistically, significant differences recorded for each amino acid as determined by t-tests are annotated with different asterisks (* *p* < 0.05, ** *p* < 0.01).

**Figure 3 plants-11-02908-f003:**
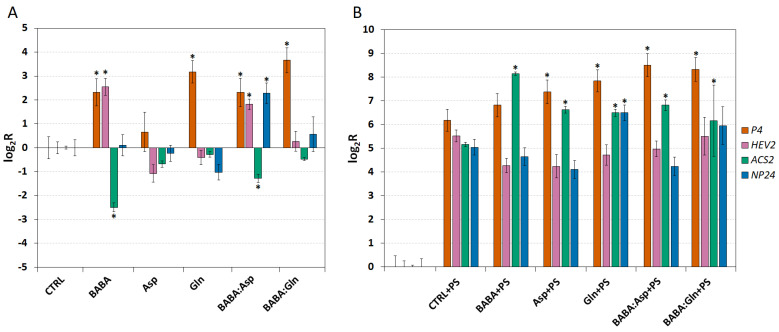
*The expression of defence genes in**Solanum lycopersicum cv. Amateur plants.* The transcript levels of four defence genes (*P4, HEV2, ACS2* and *NP24*) were monitored in 8-week-old leaflets (n = 6) 24 h after treatment with 5 mM BABA and/or amino acids solutions (BABA, L-Asp, BABA:Asp, L-Gln, BABA:Gln) (**A**) or 48 h after their inoculation with *P. syringae* pv. tomato DC3000 (**B**). Transcript levels were measured by the RT-qPCR method and evaluated by the ΔΔCt method [14]. The control tissue (CTRL) was a water-treated control sample. Each bar represents the mean ± SE. Asterisks denote mean values that differ significantly from that of the control tissue based on Dunnett’s test (* *p* < 0.05).

**Figure 4 plants-11-02908-f004:**
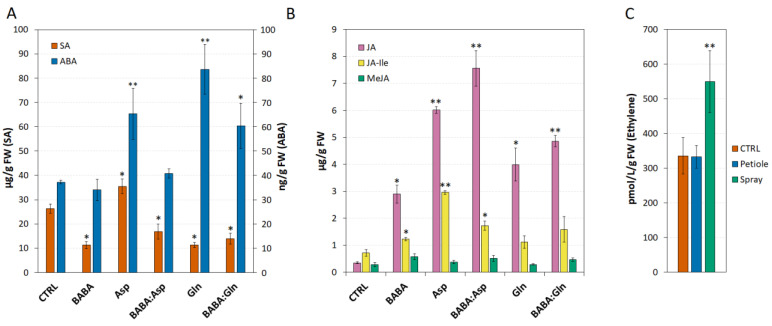
Phytohormones levels in *Solanum lycopersicum* cv. Amateur plants. (**A**,**B**) The levels of salicylic acid (SA), abscisic acid (ABA), jasmonic acid (JA), jasmonate isoleucine (JA-Ile) and methyl jasmonate (MeJA) were measured by LC–MS 24 h after 5 mM BABA and/or amino acids solutions (BABA, L-Asp, BABA:Asp, L-Gln, BABA:Gln) of the 8-week-old leaflets through petiole aspiration (n = 6). (**C**) ET accumulation was measured 24 h after BABA treatment (5 mM) of the 8-week-old leaflets through petiole aspiration (n = 6) or spraying (n = 6) by gas chromatography. The control tissue (CTRL) was a water-treated control sample. Each bar represents the mean ± SE. Asterisks denote mean values that differ significantly from the control tissue based on Dunnett’s test (* *p* < 0.05, ** *p* < 0.01).

**Figure 5 plants-11-02908-f005:**
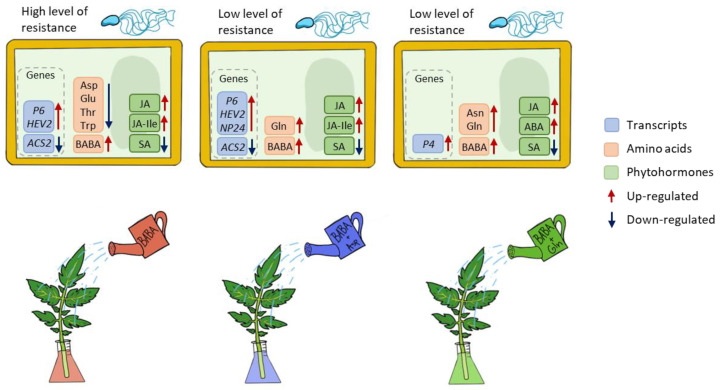
Schematic representation of processes operated in tomato plants after treatment with BABA or BABA + amino acids Asp and Gln.

## Data Availability

Not applicable.

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
