# Peer review of "L-Aspartate and L-Glutamine Inhibit Beta-Aminobutyric Acid-Induced Resistance in Tomatoes"

_plants, 2022, doi:10.3390/plants11212908_

Round 1

Reviewer 1 Report

The present study by Janotik and co-workers deals with the effect of amino acid application on the formation of beta-aminobutyric acid induced disease resistance in tomato. The study deals with a very interesting and relevant topic. 

The introduction is comprehensive and clearly written, although I think more sources should be included to further support the very clear statements. 

The effect of aspartate and glutamate on BABA induced resistance induction was studied and clearly the reducing effect. The question arises for me whether - as a control, as it were - other amino acids were also administered in parallel and it was found that these had no effect on resistance induction. If these experiments were not carried out, what makes the authors assume that this control is not useful? It is very positive that the effect was studied in both directions - the effect of Asp and glutamine on resistance induction and the effect of BABA on amino acid levels. 

The genes selected for transcript analysis seem reasonable to me and the results are comprehensible and clear, so that they support the research hypothesis. The results of the phytohormone measurement remain interesting and difficult to explain for me. It is difficult for me to get a picture of how this fits with the rest of the data. But the data is the data. 

The discussion is well written and actually helps well to better frame the results. If I could wish for something, it would be a schematic representation that graphically matches the different observed effects with possible explanations, so that one could grasp the relationships at a glance. The content of the study is quite complex and it could be helpful to have a model in graphical form. 

The methods are also well written and comprehensible. 

Reviewer 2 Report

Need to make the manuscript error-free with regards to grammar when you submit it to a journal of this impact factor. The reviewer had to edit it as a language editor.

Make sure the scientific nomenclature is consistent and right, which is a basic must for an article for peer review.

Please discuss (maybe a hypothesis/or a scientific reasonable guess) why there is a difference between the observed ET-related result when the BABA application mode is different. just merely repeating it in the paper does not add value to the discovery.

Is BABA or the amino acids in conjunction with BABA dissolved in water solvent? If not please clarify what was the solvent for these compounds.

What is the age of the tomato plants treated with BABA and the pathogen P. syringae pv. tomato DC3000 and the mode of inoculation need to be elucidated more clearly. It's very confusing to determine how exactly that was done from the current description in the methods.

Is there any information on the direct effect of BABA on Pst DC3000? If so please write a sentence or two to that effect with the required citations.

Is there a common name for the genes PSPTO_4345 and Solyc00g174340? And what is the GenBank/NCBI identifier? please clarify here.

 "The application by syringe did not induce significant resistance after the BABA or Asp treatment (Figure 1A), but application by spraying induced considerable resistance after the BABA treatment (Figure 1B)"-This brings us back to the point if BABA by itself is toxic to Pst DC? Hence, please clarification is needed if it is known, or if it was discovered by the authors of this paper. If the prior instance is there, please cite that work.

Cite here which method/paper was followed for the ΔCt method.

Line 132 - BABA treatment of both tomato cultivars did not induce a significant increase in Asp level (Figure 2A) when in the case of S. lycopersicum cv. Amateur we even detected a decrease in the L-Asp level (Figure 2B). - Please provide a reasonable explanation.

Figure 2- How many times was this three replicate/biological replicate done? One time 3 replicates, or multiple times at different time points, but each time with three technical replicates? Please clarify here.

Line160 - Always expand any acronyms at their first usage. And please redo the same when the paper is submitted with suggested revisions.

Line 172- Is BABA or the amino acids in conjunction to BABA dissolved in water solvent? If not please clarify what was the solvent for these compounds.

Round 2

Reviewer 1 Report

In the revision of the present paper, most of the criticisms have been addressed, which is why, in my view, nothing stands in the way of publication.